**National Borders among Families:  Removal and 'Bare Life' in India**

**Salah Punathil**

**University of Hyderabad**

**salahpunathil@gmail.com**

**Abstract:**

*This paper is about the state-driven process of 'migrant 'illegality' (Genova 2002) and its impact on the life of Bengali speaking residents in the Assam state of India. While the movement of people across borders between India and present-day Bangladesh has been historical and complex, this ethnographic work explores how the state-driven process of migrant illegality and the production of 'bare life' have disrupted intimate relations and family life among the migrant population in Assam. While the recent NRC (National Register of Citizens) update in Assam identified 1.9 million people as illegal migrants, there has been bureaucratic enactment of 'migrant illegality' by Assam Border Police for the last several years. The institutional procedures, court documents and narratives of the select cases of 'detected' as well as 'detained' residents  from ethnographic fieldwork reveal how the absence of formal papers and errors in the family records, kinship relations and property inheritance among the poor migrant families transforms actual citizens to 'illegal migrants' in the bureaucratic manoeuvring and reduces them to their 'bare life'. The paper also shows how prejudice, arbitrariness, and contradictions feed into the bureaucratic process and lead to intense crises among family units, as several migrant families have both Indians and alleged 'Bangladeshis' in their homes today. The paper argues that the major consequence of this state-driven 'migrant illegality' in the last two decades has been the creation of national borders among families, unsettling intimate relations and shared spaces.*

## 1. Introduction

Dugdhan Das is a middle-aged Bengali-speaking man from the Bongaigaon district in Assam, India. Having enjoyed citizenship rights all his life, he had to spend six years in the Goalpara jail before being granted bail in 2019 after the Supreme Court ordered the release of all alleged illegal migrants who had spent at least three years in a detention centre[1](Kalantry, S and Tarafder, A. 2021). He describes the process by which he ended up in jail (Dugdhan Das, excerpt from interview, 2021).

> The border police came and gave me a white paper where my name and address were written and I was asked to appear at the border police station on a specific date mentioned. On the day mentioned, I went there with all my documents. I was asked to wait there in the office. I used to run a small tea stall. In the beginning, everything appeared normal at the police station and I was hoping to return home after the police verified my documents. But later on, some ofthe activities at the police station made both of us very uncomfortable. I realised that I couldn't move without obtaining their permission. Even when I tried to go to the toilet near the police station, I saw two officers, one in civil dress and another one in uniform, preventing me from going to the toilet alone. I asked them what the matter was, but the officer almost shouted at me and said, "You are not supposed to go alone. I am sending an officer with you." I had no clue what was happening there. I totally believed that I had gone there for the purpose of verification of my documents. Meanwhile, I saw a police car arrive at the office, and some police officers started wearing their respective uniforms. After reaching the Bongaigaon SP for the final hearing, I

was asked to show all the documents that I had. The officer sitting there told me, "Now you are going to jail." I was crying and kept saying that I have three small children at home (Dugdhan Das, excerpt from interview, 2021).

In her brilliant essay, Nathalie Peutz (2006) called for an 'anthropology of removal' to understand the bureaucratic legal process and embodied experience of the migrants who are stripped of their rights and deported back to their countries of origin under heightened and stricter immigration regulations. This attention to 'removal' was an important anthropological turn prompting scholars to study 'illegality' as a socio-political and legal condition (Genova 2002; Willen 2019). 'Bare life', the concept propounded by the Italian philosopher Giorgio Agamben (1998), is one of the most influential ideas that capture the experience of removal of undocumented migrants, refugees and stateless people in the modern world. While his distinction between *zoe* and *bios*—biological and political life, respectively—is widely used to discuss the stripping of political rights from migrants in today's world, the essay explores the South Asian context of 'bare life' and contributes to the recent literature focusing on contextual and nuanced readings of 'bare life' (Fassin 2007; Ticktin 2011; Willen 2019). More historically and contextually situated analyses are interested in the dynamism of 'bare life', as scholars drawing on concepts like 'humanitarianism' (Arendt 1958) and 'biopolitics' (Foucault 1990) see it as multivalent, transitional and even a locus of mobilisation for political causes (Fassin 2007; Feldman 2015; Ticktin 2011). While the literature on 'bare life' under heightened anti-migrant regimes largely centres on the Western experience, this paper asks: What characterises the South Asian experience of 'bare life' for those who are suspected as illegal migrants ? Specifically, through a historically anchored ethnography (Willen 2019), the essay aims to unpack the

complex and gradual process of removal and production of 'bare life' among the Bengali-speaking residents[2] in Assam, an ethnic group labelled as illegal settlers because of their origins in what is now Bangladesh. The essay argues that the making of 'bare life' is an experience of gradual undoing, a process by which individuals living with citizenship rights and leading a familial life in a social setting are rendered as rightless through complex arbitrary processes and bureaucratic violence over a period of time, eventually resulting in coercive removal from their normal life. While exploring the question of 'illegality' and 'bare life', the essay also delineates how these processes are related to and influence the familial realm. In doing so, the essay attempts to understand how illegalised persons are reduced to a state of bare life and have a profound impact on their everyday familial and intimate life. Recent literature suggests that the interrelation between migration politics and the familial realm is largely undermined as most studies are confined to the multiplication of borders inside and across the nation-state and the threat of deportation in the lives of undocumented migrants. (Castañeda 2019, Van Osch 2021). However, there is little scholarly focus on how bordering works inside families as well. The recent attempt in the U.S. context focuses on how migration policies and 'deportability'are closely connected and experienced in the familial realm (See Castañeda 2019; Dreby 2012; Enriquez 2015; Hagan et al. 2008). Castañeda's brilliant work on the US-Mexican border explores the cumulative ripple effects of state policies on migration by focusing on the social unit of the family. Her works shed light on the ways in which illegality impacts opportunities for everyone in the familial setting, as individuals are always embedded within complex social units. This essay adds to this emergent body of literature by highlighting the specific experience of migration politics in India.

The historical context of cross-border migration is crucial to understanding the specific Indian scenario of migrant 'illegality' and the experience of 'bare life'. The alleged illegal migrants from Bangladesh are those who crossed the borders after 24 March 1971. But Assam has a much longer history of movement by Bengali-speaking populations, who have arrived in several waves from what is now Bangladesh since the late 18th century (Baruah 2007; Gohain 1985; Guha 2014; Hussain 1993; Punathil 2021). Factors such as overpopulation, poor natural resource bases, frequent floods, non-diverse economies, feudalism and overreliance on jute and rice have historically pushed people to make this move (Baruah 2007; Hussain 1993; Punathil 2021; Weiner 1983). After the British annexed Assam in 1826, they encouraged such migration as part of plans to plunder and exploit the new territory (Gohain 1985; Guha 2014; Hussain 1993). Although both India's Partition in 1947 and the Bangladesh War of Independence in 1971 further accelerated the movement of people across borders, a large chunk of the population living in Assam are Indian citizens whose ancestors settled there long before the formation of nation-states in South Asia. However, for the past five decades or so, the Bengali-speaking population is invariably 'migranticized' (Dahinden 2016) and labelled as a threat population and is being accused of the following, among other things: altering Assam's demography, encroaching on the land of native communities, taking resources and economic opportunities away from local people, forging documents such as electoral cards, influencing local politics and even threatening Assam's culture (Gohain 1985; Guha 2014; Hussain 1993).

A few studies have shown the importance of understanding emergent and complex socio-political and economic practices that transcend national borders (Chowdhury 2020; Ghosh 2023; Schendel 2004; Sur 2021). There have been trade relations across the borders of India and

Bangladesh, including rice and cattle, that often come in conflict with state policies and border security forces owing to illicit practices and yet managed through informal means (Schendel 2004; Sur 2021). Migrants from Bangladesh constitute a huge labour force in Assam and elsewhere in the country as their fragile social location offers cheap labour to various economic sectors (Gandhi 2017; Misra 2018). Marriage and kinship relations have been established across borders along the mobility of humans and the movement of material goods (Ghosh 2019; Ibrahim 2021). Although the Bengali-speaking population in Assam now constitutes both citizens who settled or have ancestors in India before 1971 and those who crossed illegally, they have not been differentiated or put to the test in citizenship terms until the 1990s[3].

The academic discourse on migration in Assam has been primarily centred around the question of the illegal acquisition of citizenship rights and its consequences in the state (Punathil, 2022). In such literature, the migration in Assam is posited as a case of unique challenges in studying the movements of peoples across borders in South Asia (See Baruah 2009; Sadiq 2008 and others). Scholars such as Sadiq (2008) and Baruah (2009) contextualise the Assam scenario in light of a specific strand of migration literature that argues that migrants in the West are integrated to meet their economic needs primarily but are deprived of basic citizenship rights, which creates a demarcation between citizens and non-citizens. In contrast, they argue, there is an inherent difficulty in distinguishing between citizens and illegal migrants in India. Scholars, thus, advocating 'indistinguishability' argue that illegal migrants have been acquiring documentary citizenship fraudulently after crossing the porous border between India and Bangladesh (Sadiq 2008; Baruah 2007). Sadiq argues his case with empirical data drawn from official sources, records, statistics, and political narratives and states that a huge number of

illegal migrants enjoy the benefits of citizenship rights, including voting rights. His study looks at Indonesians and Filipinos in Malaysia, as well as Afghans and Bangladeshis in Pakistan. However, a major portion of his work deals with the issue of migration from Bangladesh to India, especially Assam. While the above-mentioned 'indistinguishability' discourse is largely centred around the methodological nationalism (Mongia 2018) that predominates the discourse on migration in Assam, little has been said about how discursive practices around 'illegal' migration have removed people from their social world and rendered them as 'bare life'.

## 2. Methodology

The methodological challenge in researching the removal is the invisibility and extreme vulnerability of the subjects, be it the left-out people from NRC, D voters, referred cases, or the detained lives in jails. D voter is a category of voters in Assam who are disenfranchised by the government on account of their alleged lack of proper citizenship credentials. Referred cases are instances of possible illegal migrants to be surveyed in a cluster of villages under the jurisdiction of the border police. They are neither declared illegal immigrants nor do they enjoy citizenship rights. There is no access to the detained kept in various jails of Assam; those who are suspected of being illegal migrants and failed to prove their citizenship in NRC often hide from public life as they are under perpetual threat of detention and deportation. However, this research has been facilitated by the emergent activists within the community who mediated between the researcher and the subjects. The insights presented in this essay are the outcome of the fieldwork I have carried out in Assam at regular intervals since 2019. I visited the field site at regular intervals until the year 2022; each visit consisted of a few days of stay in the localities inhabited by the Bengali-speaking population in the Barpeta district of Assam. The fieldwork gave special

emphasis to the life stories of the detained people who got released from detention centres after months and years of staying there. I had long conversations with five detained people apart from their family members and the D voters. The study entails the ruptured stories of those who live in an anguished world, those who lost citizenship rights and lived with precarious citizenship. This includes those who got temporarily released from detention centres and want their stories to be heard by the world. The homes of the detained turned out as the prime field site of this work as long conversations with the detained and their family members unravelled the ways in which the state-driven illegalisation process irreparably damaged the lives of people.

Interviews with detained individuals largely unveil a trajectory of life from the status of citizenship to the condition of partial rights after detection by the state and complete rightlessness after detention. There are two categories among the detainees: those who got released after a legal fight proving their citizenship and those who got released after a Supreme Court verdict in 2019, which mandated the release of people who had been detained for more than three years. The homes of the detained are emblematic of various forms of vulnerabilities, especially the perennial effect of state intrusion into their family life. Since the study aimed to look at how the individuals and their entire families are affected in the context of detection, detention and deportability, special attention has been given to the aspect of how family members have experienced the removal of a member from their day-to-day life. The narrative of family members yields how detention made their lives precarious as the pain of separation, economic burden and an overall fissure in the familial life pushed them into a calamitous zone. Apart from narratives of the detected, the detained, and their family members, the data includes the judicial documents, petitions and other files pertaining to the cases presented in the

'Foreigners Tribunal'. This has been especially useful in understanding the ways in which legal and bureaucratic interventions strip the rights of people in Assam.

## 3. Conceptual Issues

This section explores the conceptual aspects that help to understand the dynamics of the relationship between the political process of detention, detection and deportability, and the familial and intimate space. The article primarily draws on the concept of 'bare life' as propounded by Agamben. His core argument rests on the premise that in modern sovereign states, especially during times of 'exception', individuals can be reduced to a state of 'bare life'. It is a life in sheer biological form, stripped of any legal and political significance, where one exists in a zone of 'indistinction' between the political and the biological (Weber, 2012). The state of exception is not a transitory or an exceptional anomaly but can be normalised in modern political structures. Within this very apparatus, the zone of indistinction proliferates, perpetuating the obscuring of *bios* and *zoe*. It is a zone of indistinction because as the difference between *bios* and *zoe* gets blurred, the individuals are reduced to their mere biological existence, deprived of the rights and protections that are otherwise associated with political life (Owens, 2010). The complex scenario in Indian context demand a more nuanced reading of Agamben. There are spectacle instances and massive production of 'bare life' like the NRC list in 2019 in which a large number of people were stripped of their citizenship status. The citizenship crisis has led to an alarming situation for these individuals where they find themselves in a constant state of precocity, facing the risk of becoming stateless and/or losing their legal and political rights and being in a condition of politico-legal limbo. The entire process of the NRC ends up being a process of exclusion that can render individuals politically invisible and marginalised in the

realm of everyday life. However, this essay bring the stories of those who have been subjected to detection and detention much before NRC implementation as there has always been legal process and bureaucratic- legal interventions that led to the extreme suffering of Bengali speaking residents. In the new scenario of NRC and in the earlier cases, those lived with citizenship rights, are subject to gradual irregularisation that reduces them to their bare life as they are illegalised and removed from the social world. In this way, the essay move beyond Agamben's spectacle notion of 'bare life' that there is a gradual process of undoing the citizenship. More importantly, this essay nuances 'bare life' ethnographically as it interconnect the political and the familial and extrapolate the intricate connection among them.

In this context, the concepts of 'intimate citizenship' and 'irregular citizenship' are useful anchors for a new reading of the complex citizenship process in India. Intimate citizenship as a concept is originally associated with studies on family, gender and queer studies, and disability studies. The idea was first explored by Kenneth Plummer in the 1990s, and he published it as a book in 2003 (Plummer, 2003). The concept of intimate citizenship, though explored in the studies on family policies, sexuality and disability, has gained new momentum in the recent debates on migration, law and ethnicity in the Western context (See Bonjour and Hart 2021, 8; Odasso 2021, 76; Hart and Besselsen, 2021, 38; Griffiths 2021, 21). In the current context of citizenship regimes, the idea of intimate citizenship is being explored to look at how the intimate familial life is being affected due to migration laws and migration policies of liberal democratic states. In the recent volume of the journal 'Identities: Global Studies in Culture and Power' (2021, Vol 28), scholars have paid attention to a variety of issues pertaining to mixed-status families and the particular effects of migration policies on these families. Intimate

citizenship employs an intersectional lens and puts emphasis on how essentially 'private' concerns have a lasting impact on issues pertaining to citizenship. In the process, it is possible to investigate citizenship concerns as well as personal and everyday crises that invariably address the claims of belonging. Thus, intimate citizenship views citizenship as an embodied practice and everyday experience (Lister, 2007). In his work, Kalir (2020) has shown the influence of kinship on migration policies in state-making and the affective fashioning of national belonging. Castañeda (2019) has argued that illegality and deportability are constituted and reconfigured through intimate relationships. In other words, intimacies are material sites that are always connected to larger relations of power and governance **(**Castañeda 2019**)**.

Another conceptual framework that has guided this essay is the idea of irregular citizenship, which delineates the ambiguity and messy nature of the evolving status of migrant populations under definite socio-political and legal conditions (Nyers 2019, 21; Isin 2009, 217; Squire 2011, 4). Irregularity can point out a range of things that often include the status of ambiguity, the incongruous experiences of citizenship and non-citizenship and the potent threat of removal and deportation from a nation-state (Nyers 2009, 188). These two concepts are also closely linked to the idea of precarious citizenship, as irregular citizenship and intimate citizenship are about the precarious situation of people that emerges from the struggle to gain access to resources and to enjoy the full benefits of citizenship rights in a nation-state (Gold-ring et al. 2009, 245; Lori 2017, 17; Parla 2019, 21). What defines precariousness is the sheer state of vulnerability, unpredictability and insecurity and the ones experiencing this predicament are prone to the risks of poverty, disease, displacement and extreme forms of violence (Lori 201, 9). The idea of precarious citizenship attempts to encapsulate the arbitrary, messy and fragile situations where state-

driven interventions push a group of people into uncertainty. It also provides a space to unsettle the binaries of citizenship/non-citizenship and insider/outsider, as these complex experiences, more often than not, involve all of these ideas at the same time (Goldring and Landolt 2021, 2; Lori 2017, 8; Ramirez et al. 2021, 23).

Connecting 'bare life' to the ideas of irregular and intimate citizenship provides an interesting framework to analyse how individuals who have been gradually irregularised and stripped of their citizenship status exist in a state of ambiguity on an everyday basis. As a result, the intimate familial life is pushed into jeopardy, where crisis grips the members. Thus, family as a unit of intimate life can be a useful lens to analyse 'bare life' as it becomes a witness to the ramifications of arbitrary political interventions of the state. By extending this debate to the South Asian context, this essay aims to explore two interrelated themes of intimate citizenship - a) How family itself becomes a quintessential unit of defining citizenship in the policies of the state and how it affects and irregularises citizenship status of individuals; b) How citizenship policies and bureaucratic interventions produce mixed families having both Indians and alleged Bangladeshis at their home, leading to an intense crisis in their life.

### 4. The Process of Irregularisation

A new National Register of Citizens (NRC) in 2019 saw the listing of 1.9 million people as illegal migrants in India's northeast state of Assam before the Citizenship (Amendment) Act was passed**.** NRC is a bureaucratic solution to the long-standing demand for detecting illegal migrants, and this has to be seen as an exceptional state mechanism in India in order to reinforce territorial boundaries through strict control over the migrant population as seen in the other nation-states globally (Roy 2016). The NRC was implemented to create a register of citizens

based on the 2003 amendment of the Citizenship Act of 1955.  In the Supreme Court order of

2014[4], it was stated that the Supreme Court would directly supervise the progress of the entire

update of the NRC facilitated by the BJP government both at the centre and at the state level.

While the recent NRC update in Assam identified a large chunk among the Bengali-speaking

population as illegal migrants, there has been bureaucratic enactment of 'migrant illegality'

through 'D' voters or doubtful voters list and detection of 'illegal migrants' through 'referred

cases' by Assam Border Police for last two decades. Several detected migrants have been sent to

various detention centres during these years.

The process of irregularisation is in practice two-fold – while citizens are irregularised by having

no claim to the rights that they are rightfully owed, irregularisation also occurs through the

various paths taken by citizens and non-citizens in order to acquire rights and transform the

boundaries of belonging (Nyers 2019). This process of irregularisation is not narrowly confined

to the categories of refugees, asylum-seekers, migrant workers, temporary residents, or

undocumented migrants anymore. The tendency of hardening citizenship policies involves a

stricter formalisation of documentary procedures and a tightening of control practices; on the

other hand, such moves have increasingly led to forging more documents through illegal ways

(Chauvin and Garcés-Mascareñas 2012). In fact, citizenship is continuously being made and

unmade through this process of irregularisation. In other words, the process of irregularisation is

what makes or unmakes people into irregular citizens. The process of irregularisation of citizens

in Assam owes to the post-colonial history of ambivalent legal interventions, ineffective policies

and inconsistent bureaucratic practices (Tuckett 2015, 115). It was the Assam Accord formulated

in 1985[5] first offered a concrete solution to detect the alleged 'Bangladeshi migrants' in India

(Baruah 2009; Ranjan 2019; Sharma 2019) as it demarcated 25 March 1971 as the cut-off date to differentiate citizens and non-citizens in Assam. The Illegal Migrants (Determination by Tribunal) (IMDT) Act of 1983, passed in 1983, insisted on setting up special tribunals to examine the cases raised by ordinary citizens and police regarding people suspected of being illegal migrants (Jayal 2013, 65; Roy 2016, 46). The process of detection under this act proved to be ambivalent and largely futile. The obligation of evidencing the illegality of a migrant rested on the accuser and someone who is residing within a three-kilometre radius of the alleged illegal migrant. The accuser also needs to furnish corroborating affidavits by two more persons who are also residents within that radius (Ranjan 2019, 448; Sharma 2019, 533). The act has to be read as a tactic of the performative state to pacify the political turmoil that has been embroiling Assam since 1971 to detect illegal migrants, as this act is found to be ineffective in detecting illegal migrants. The irregular citizenship in Assam has its roots in such blemished, indeterminant and double-edged legal interventions as such policies reinforce the ambiguities over citizenship (See, e.g., De Genova 2016, 94; Sur 2021, 227; Tuckett 2015, 116) Moreover, people have access to fraudulent government documents through a corrupt low-level bureaucracy in post-colonial states like India (Gupta 2012, 33) or through various informal networks, irrespective of their actual status. This 'blurred membership' (Sadiq 2008) makes it hard to differentiate citizens and non-citizens as people are either permitted to be invisible in 'real' practice, in opposition to 'official' norms (Tucker 2015, 114) or prompted to obtain elementary paper credentials by illicit means.

In the Assam context, irregularity is no longer simply about how non-citizens fall prey to irregu-larising practices but also about how citizens are subject to irregularising practices and made into

irregular citizens. This has far-reaching consequences on the status of those who live with such citizenship credentials, especially when a strict policy of state comes into effect (Chauvin and Garcés-Mascareñas 2012, 254). Irregular citizenship is, therefore, the outcome of the long-term policies of the state (Gupta 2012, 6; Hull 2012, 253). While the IMDT Act is the first attempt to irregularise the citizen, it is the establishment of the category of 'D voter' that made ordinary citizens into irregular citizens. Studies (Nyers 2019) have shown that irregularisation does not necessarily mean that their citizenship status is revoked or blatantly taken away. Instead, what is more appalling are the ways in which the citizenship of an individual is rendered non-functional or often inoperable in certain discrete situations. The context and processes involved in the process of irregularisation need to be critically studied and analysed as these are highly contested. 'D voter' (doubtful voter) in Assam is about disenfranchised people who are suspected of being illegal migrants in the absence of proper citizenship credentials. It was in the year 1997 that the Election Commission of India declared more than 100,000 people as D voters (Sharma 2019). Another state intervention that led to the irregular citizenship in Assam is the establishment of the 'reference case'. The border police force in the concerned jurisdiction is entrusted to survey a cluster of villages to detect illegal migrants, which are regarded as referred cases. There have been a large number of people living under this category over the past two decades with a status that neither caters to being a citizen nor an illegal migrant (Heath Cabot (2012). In 2005, the Supreme Court scrapped the IMDT Act after a series of legal contestations and deliberations between various stakeholders[6]. Now, the onus of proving citizenship is on the person who is accused as an illegal migrant. Since 2005, there have been numerous cases reported and processed

at the Foreigners Tribunals in Assam, leading to the illegalisation of the Bengali-speaking population.

## 5. Intimacy and Citizenship

This section looks at the ways in which incongruence between state devices of identifying citizens or illegal migrants and the complexities in the real-life situations of intimate relationships impact the process of irregularisation and illegalisation of citizens in Assam. The analytical framework of intimate citizenship helps in shedding light on how citizenship is tied to intimate life (Bonjour and Hart 2021; Odasso 2021; Hart and Besselsen 2021; Griffiths 2021). Exclusionary border controls, restrictive migration policies, restrictive admission and restrictive deportation are all at the heart of the intimate citizenship framework, as they are invariably linked to the unit of the family. The policies adopted to determine the citizenship status in Assam have been favouring jus sanguinis (law of blood-based citizenship), hence considering the family as a unit to define the citizenship status. The recent NRC asks every individual to present Legacy Data to prove their relationship with ancestors who lived in Assam before 24 March 1971, the cut-off date to consider whether one is legal or illegal. The NRC takes into consideration the Family Tree as the fundamental premise of determining citizenship and/or non-citizenship. The various details present in the Family Tree include the different generations of one's family comprising the names of the Legacy Person(s) and their children and grandchildren. The format of the Family Tree is designed by the NRC authorities to collect the family details of the applicants. To be considered a citizen, one has to prove that they lived or came to India before 1971. Those who are born after 1971 have to show that their ancestors lived in India prior to the cut-off year.

Those who were excluded from the NRC in India are those who failed to establish their ancestral roots. This is largely due to the absence of proper documents, bureaucratic errors and other discrepancies related to paper records on familial relationships. This holds true for D voters and referred cases, as I mentioned above. By citing specific instances, I argue how incongruencies between bureaucratic principles and complex family history in real life led to the undoing of citizenship rights of people.

Sharu Sheikh, a Muslim man from the Barpeta district, was a referred case in 2002 (See Civil Extra-Ordinary Jurisdiction, 2016).[7] No measures were taken against him for a prolonged period of time. However, like many others, he received a letter from the Foreigners Tribunal and was eventually sent to the detention centre in the Goalpara jail. He remained there until 2019, when the Supreme Court issued an order that allowed him to return home. His case perfectly shows the irregular yet prejudiced practices of bureaucracy and how a citizen's life is transformed into precarity (Tuckett 2015, 115). While there was no mention of the specific grounds on which he was suspected as an illegal migrant, he carefully furnished all the necessary documents to the Foreigners Tribunals in an attempt to prove his Indian citizenship. However, he was declared a foreigner because of contradictions and lack of proper evidence in his legacy data. His peculiar family history made Sharu Sheikh a 'Bangladeshi' in the eyes of the state. He was born to the second wife of his father, Fetu Sheikh, who had remarried after the passing away of his first wife. He had six children during his first marriage and his name appears alongside his first wife and his six children in the National Register of Citizens in 1951. Again, his name appears alongside Sharu's mother, Jhakani Bewa, and his elder brother, Baru Sheikh, in the electoral rolls for 1966 and 1970. Sharu was still not enlisted in the electoral rolls and when he was finally

enlisted in 1985, his father's name was wrongly recorded as Fetu Choukidar (as he worked as a night *choukidar* or a watchman for a government office). To add to further inconsistencies, Fetu Sheikh had moved twice as a result of flooding, and thus, his land records showed discrepancies. The consequence of it all was faced by Sharu Sheikh when he was declared an illegal migrant and left to suffer in a detention centre for five years. This exemplifies what Castañeda (2019) calls 'bureaucratic disentailment', a process by which administrative agencies deprive individuals of their legal position as citizens and infringe on their rights and subject them to extreme forms of vulnerabilities.

The process of irregularisation and illegalisation has a gendered impact, and cases from the field show how marriage and incongruencies in family records make women more vulnerable to state actions. Studies on intimate citizenship have shown that the vulnerabilities are tougher for women as they have little or no agency to pass on their citizenship rights, which also shows the gendered impact of restrictive migration policies and citizenship laws across the globe (Lister, 2002). In order to prove their eligibility to be included in the NRC, those who are presently married and part of other families are required to furnish the legacy data of their own parent's ancestors. This creates further complications, as the instances below show. Mariom Bibi, 43, was first categorised as a D voter in the year of 1997 for not having legacy data to prove her citizenship[8]. In 2014, more than 14 years later, the Foreigner's Tribunal sent her a notice stating that she was suspected of being a foreigner. Mariom Bibi has gone through a lot of hardships to prove herself as an Indian due to her peculiar family history and marriage. She was born in the Bodo tribal community, one of the earlier settlers and indigenous tribes in Assam, and her name was Ladhuri Bala Sutradhar. She first married a man from her own tribal community. Her

husband died, and later, she married a Muslim man named Kafil Ali and converted to Islam. Her new name, Mariom Bibi, is not found anywhere in the old records, and the officials declared her an illegal migrant. Her present name and identity as a Bengali-speaking Muslim, a community largely suspected of illegal migrants from Bangladesh, made it easy for bureaucrats to assume and declare her as an illegal migrant (Mariom Bibi, excerpt from interview, 2021). There are several cases where complications arising out of marriage become a key factor in losing citizenship rights, especially among Muslims. Since married women replace their maiden names with their husbands' surnames, they often fail to furnish the required documents that prove the relationship with their maternal and paternal families. For example, it is a common practice among Muslim women to change their second name from '*Khatun*' to '*Begum*' upon marriage. In the recent NRC, discrepancies in names, including spelling errors, have negatively impacted how the provenance of paternal family linkages is established; this led to the exclusion of many women from the list. Since girl children hardly get access to education and women do not own land, required credentials of educational certificates and land records are rare among women. Narratives from the field reveal how child marriage among Muslims positions women in a disadvantaged position to prove their citizenship. Since child marriage is legally invalid, there is a practice of wrongly showcasing a higher age for girls on the marriage certificate. This record then comes in conflict with the age mentioned in other documents and eventually leads to the dismissal of their credential of the citizen. A woman has little agency with regard to the impact she has to shoulder as a part of state policies and family practices. The examples show that there is a clear intersection between the politics of citizenship and belongingness and practices of family and intimacy. Intimate citizenship shows how personal choices often come in conflict

with the interests of the state and are almost always detrimental to the illegal 'other' who has no privilege of making claims to citizenship rights.

## 6. The Experience of Removal

This section looks at how citizens are removed from their social and familial world and coercively pushed to the dark zone of detention centres. There are six detention centres in Assam located inside usual jails; these are temporary arrangements to accommodate the alleged migrants. There are more than a thousand people staying in those detention centres at present (Nazimuddin 2020). Dugdhan Das, the middle-aged Bengali-speaking man I mentioned at the beginning of the essay, narrates:

> I kept telling them I was an Indian citizen; it was just that I missed attending the 17 court procedures to submit documents earlier due to the accident I had in between. But one of the officers said, "We have no option left. If possible, talk to your relatives to write an application to reduce your punishment". I even could not inform my family that I was going to jail. We were taken to the nearest hospital for my medical check-up and then straight away sent to jail. When I entered the jail, I got some basic stuff like one mosquito net, three blankets, one plate and one glass. While I was eating my first meal inside the jail, I felt that I wouldn't survive long inside the small room where I was kept along with many other criminals. The next morning, we were asked to stand in a queue, and they started counting us; we already had our respective identification numbers. Jail attendants asked us to keep our belongings in cell number seven and eight, respectively, cell number eight for me (Dugdhan Das, excerpt from interview, 2021).

Sharu Sheikh describes his experience:

One day, I went to the regular hearing of my case, where I got to know that I would be taken to the Goalpara Jail. They declared me a foreigner or Bangladeshi and took me. So, after a long legal battle from 2011 to 2016, they took me to jail. My family members got to know the place of my detention only after three days. Finally, my family members came to see me after four days. In these four days, I did not take a bath; even the clothes remained the same. Then I got some clothes from home (Sharu Sheikh, excerpt from interview, 2020).

The practice of keeping alleged migrants in jails along with criminal prisoners is a clear violation of United Nations Human Rights Council regulations, which insist that 'states are obligated to place asylum-seekers or immigrants in premises separate from those persons imprisoned under criminal 18 law' (UN General Assembly 2008: 20). The crux of the UNHRC guidelines is that there is a visible difference between detention of alleged migrants and punitive nature of prisoners. Until very recently, the alleged migrants were kept in jails for an indefinite period in Assam. UNHCR guidelines state that the detention cannot be prolonged unless it is absolutely required and there are reasonable conditions and legitimate purpose (UNHCR 2012). Many are forced to stay for a prolonged period of time without being provided a fair chance to prove their nationality. In Assam, there is no apparent difference in the detention of alleged migrants and imprisonment since both take place in the same jail. However, a migrant's life inside the jail is worse. This situation indicates that detention must be located within the nexus of diverse forms of captivity and confinement under sovereign power (Genova 2022). The police, the bureaucrats and the jail administrators symbolise the sovereign where the brutality of the state is nakedly exposed (Agamben 1998). Here, it is important to cite Hannah Arendt as she brought to the fore

the irony of Nazi Germany, where a common criminal possessed more legal rights and recognition than those kept in the Nazi concentration camps or those who were relegated to the status/condition of stateless refugees (Arendt 1958). The crucial point is that criminals are subjected to the law and, consequently, the punishment rules of a state. In contrast, a detainee is subjected to an administrative apparatus rather than the law itself; hence, the detainee features outside the purview of the law altogether. This is a sheer paradox of the situation where the people are first stripped of their legal and political rights, preventing them from seeking legal remedy as they are no longer lawful citizens. There are striking differences in privileges between a detainee and a prison convict. Unlike the prison convicts, a detainee cannot work or earn any wage. A prison convict is still considered a citizen, whereas a migrant is stripped of their citizenship rights. Lack of opportunities for earning poses a hindrance to the well-being of their family who lives outside. There are no recreational facilities for detainees, which makes their lives mundane and monotonously painful. On a daily basis, they wake up early and stand for the counting procedure to ensure their attendance. Next, they have breakfast and, subsequently, lunch, after which they go inside their wards upon having an early dinner at around 4 PM. There are no sources of entertainment like newspapers, libraries or television, unlike the jail cells for criminals. Detention cells are majorly overcrowded, like in the Goalpara jail, which hosts 439 detainees instead of the regular capacity of 370 people (NCHR Report 2018). What distinguishes alleged migrants from criminal prisoners is that the migrants are guilty of their status as 'illegal migrants' - they are simply penalised for being who and what they are and not for any act of wrongdoing (Genova 2002). In many countries, detainees are escorted by staff to visit their family and community in critical situations of crisis, such as illness and death or to attend their

funerals. The scenario in India clearly violates the guidelines given by UNHCR (UNHCR 2012) as it insists that upon the detainee's request, a migrant detainee should be allowed to meet the family needs, such as making phone calls, allowing to see them in prison and giving permission to meet them in crisis such as death in the family.

## 7. Impact of Borders Inside the Family and Community

In the Western context, the question of intimate citizenship primarily deals with mixed-status families that provide novel insight into how citizenship is essentially a lived practice that shapes and is, in turn, shaped by meaningful social relationships in general and family relationships in particular (See Castañeda 2019; Dreby 2012; Enriquez 2015; Hagan et al. 2016). In the Indian context, mixed families are a product of discursive and governmental practices of the state as the process of detection and identification of alleged illegal migrants turns a normal family into a mixed one. Mixed-status families then live in continuous vulnerability and anxiety with the constant fear of being separated, abandoned and deported. Interviews with D voters, residents categorised as 'referred migrants' and detained people give a sense of how national borders are drawn in the realm of social and community life as an effect of bureaucratic enactment of migrant illegality. When I spoke to Asma Khatun, an elderly Muslim woman, I learned how the tag of 'foreigner' impacts the marriage proposals of young girls in such families. Asma Khatun's husband has been detained for two years, and for this reason, their daughter's marriage proposals are turned down as there is a stigma within the community itself to not engage in relationships with families that are having suspected migrants. This crisis is not typical to the family of the detained alone; the D voter's family is also not the first choice for those looking for a marriage

alliance. Abdul Salam, a D voter, tells his experience when he tried to find a suitable alliance for his daughter.

> Recently, everything was finalised about her marriage as the bridegroom and his family liked her and us. People from our side went to the groom's home for religious prayer and to fix the date for marriage. After things progressed, someone told them that the girl's father was a D voter. Then, they rejected my daughter. We spent around six to seven thousand rupees on marriage-related activities already, and I can't look at my daughter's face now (Abdul Salam, excerpt from interview, 2021).

Abdul Salam says that they treat many families with food and great hospitality when they come to see their daughter, but every proposal eventually gets rejected when they realise that he is a D voter. Like Abdul Salam, many of these families prefer not to reveal their status as suspected citizens to families who come with the proposal since it is a negative marker, but this 'secret' is discovered when neighbours or local people inform the family who comes with a proposal. In the recent past, marriage alliances among Muslims in Assam were fixed only after verifying that there were no suspected or detained individuals in the family.

Once a person is detained, it disintegrates the entire family. In most cases, it is one or two members of the family who are declared as foreigners, whereas the remaining others are Indian nationals, and this situation is evidentially illogical. There are many cases where the parents are declared as foreigners while children are Indians and on the other hand, children are declared as foreigners whereas parents are Indian citizens. This is true in the case of several families that have been left out of the NRC. Hence, while migrant detainees undergo intense suffering inside

the detention centre, their remaining family members living in rural areas in Assam experience trauma and uncertainty about their kin's future. A detained migrant can keep only his or her children below six years along with the detained in jail. The absence of a parole system leaves no chance to be with family after detention. Although family members are allowed to go and meet the detained person, it is not practical for many families as it is expensive to travel long distances. Until 2014, there were only two detention centres in Assam. The detainees whose families reside very far away and are unable to visit them in jail frequently are not even allowed to communicate over the phone with the detained. Children, after completing six years in jail with their parents, are supposed to leave the jail. In such situations, the state does not take any responsibility for the child and the legal provision is very unclear. In most instances, distant family members take care of them. Many detained  Bengali speaking women in Kokrajhar jail are having their small children with them and they are sent to a primary school near the jail. Shahera talks about this while narrating her experience - "I have seen jail authority providing school facilities for those children. Police used to take responsibility for transporting children from jail to school and back. Expenses are taken care of only by the government authority. They provide school books, tiffin, uniforms, etc." (Shahera Khatun, excerpt from interview, 2020).

Momiran Nessa had three kids while she was sent to the detention centre. Her elder daughter was 12 years old, and her two sons were six and three years old, respectively. She has not taken even the three-year-old child along, thinking that the kid will have a better life at home than living with her in the detention centre. Momiran was about to cry while recalling the pain of separation. It is striking to notice that the relatives of most detainees are excluded from the recent NRC list in certain cases and hence became illegal migrants because of the status of the detained migrant

as 'illegal'. Momiran says, "Because of my identity issue, names of other family members did not appear on the NRC list. It is very unfortunate to see my entire family suffering because I am detained" (Momiran Nessa, excerpt from interview, 2020). Moslem Ali, the husband of Mozira Khatun, who is detained in the Goalpara detention centre, narrates, "Usually when parents die, children become orphans. But I feel that I have become an orphan since my wife was detained. It feels like she is dead even when she is alive. Who takes care of the family now?" (Moslem Ali, excerpt from interview, 2021). These narratives depict how deep-rooted the impact of such arbitrary legal detection and identification of individuals can be as a result of which the normalcy of everyday familial life gets jolted.

## 8. Precarious Lives

Life in a detention centre reveals the power of the state to put the migrant's life on hold indeterminately (Hull 2012; Hasselberg 2016). The detained  Bengali speaking residents in Assam are subject to banal administrative power; the indefinite waiting and protracted uncertainty under torturous conditions in jail shape their lives as "terribly and terrifyingly normal" (Genova, 2002). The right to parole is only reserved for prisoners; alleged migrants are not allowed because they are not considered Indians in the eye of the state. Migrant detainees are denied exit in crisis situations like death in their family. The denial of parole and even permission to visit family in circumstances like death adds to the deplorable condition of migrant detainees as incarceration on the basis of 'migrant illegality' and leaves them without any legal remedy or appeal. Alleged migrants hardly find a legal option to escape from the present condition of being kept in jail.

The medical negligence of the detainees is particularly striking here. The stories of those who lived in the detention centre reveal that the elements of compassion and humanitarianism are completely missing in the Indian context. Dugdhan Das had developed a cataract problem in his eyes after he spent one and a half years in jail. Once he realised he was losing sight of one eye, he asked the jail authorities for a check-up. After his repeated reminders, he got the doctor to consult and diagnose the disease. Initially, he got medicine, but as the illness prolonged, the doctor suggested going for surgery. However, it took one year for him to get his surgery from the Goalpara Civil Hospital. In between, he was undergoing cloud vision and desperately looking for his treatment. Once his surgery was done, he got glasses and everything looked alright. However, he soon realised that his second eye was undergoing the same problem of cloud vision. This time, it was even more difficult to convince the jail authorities. He narrates:

> Though I was taken to the hospital a few times, the surgery for my second eye has never happened. I was unable to understand what was happening to my case as I had little opportunity to pursue the surgery when locked up in a detention centre. My problem was simply neglected. I got released a few months back. Now, I don't have financial backup to do the surgery for my second eye; my overall health has deteriorated and I am very weak now. My family is unable to find a livelihood. I started doing some manual labour, but I can't work for long. I cannot do any work…Every 30 minutes, I need a break while working. I can overcome my emotional trouble, but this eye, I can't help. I don't have money to fight the case; I am also worried that they may catch me again and put me in the detention centre (Dugdhan Das, excerpt from interview, 2021).

While few actual illegal migrants in the detention centre do not contest the state and prefer deportation to Bangladesh, those who contest the state-sanctioned status of 'illegal migrants' in Assam can never imagine deportation as it will be removal from their family forever. In the eyes of the state, deportation is about sending back the 'unwanted' to Bangladesh and stripping them of the citizenship privileges they enjoyed. But it is a complete underestimation of the sociality and intimate relations that have emerged over decades in a region where the social world has been constituted across and beyond the borders. The stripping of citizenship is about the disruption of the intimate life of individuals. As mentioned earlier, it was a Supreme Court order that enforced the existing rule that a migrant detainee cannot be kept in a detention centre for more than three years. To get released from jail, the detained migrant has to furnish a bond of Rs. 1 lakh with two Indian citizens and give details of their address of stay after release. The stories of such released individuals  unravel the terrible impact of detention on the life of Bengali speaking residents as the rhythm of the migrant's life is irreparably damaged during the detention period. It is the unfeasibility of deportation that led to the release of detainees and they are never free from the fear of detention and deportation. A released detainee's life is profoundly shaken, and his or her life would never go back to the earlier stage. Detention is a separation from all material and practical life she or he has been nurturing and sustaining for years.

Once a suspected person receives an official notification or is detained, a huge financial challenge confronts their family, from finding a lawyer to furnishing certificates, attending the Foreigners Tribunal and winning the case. Some sell their property because most of the best lawyers charge high fees. Without a competent lawyer, the chance of losing the legal fight is very high, but since suspected migrants invariably lead precarious lives, most can afford only a less expen-

sive lawyer with whom they must place their complete trust. However, many lawyers simply take the money and perform their duties without adequate effort. While reading the arguments presented at Foreigners Tribunals, I clearly saw that the lawyers are often ignorant of the legal complications and nuances of their cases and simply make rhetorical statements on behalf of the poor and illiterate petitioner instead of introducing favourable facts and evidence. Sharu Sheikh, a Muslim man from the Barpeta district, was a referred case in 2002 (See Civil Extra-Ordinary Jurisdiction, 2016). No measures were taken against him for a prolonged period of time. However, like many others, he received a letter from the Foreigners Tribunal and was eventually sent to the detention centre in the Goalpara jail. He remained there until 2019, when the Supreme Court issued an order that allowed him to return home. When Sharu Sheikh was already in a disadvantageous position because of the incongruities and errors in his documentary evidence, his lawyer made contradictory statements in the appeal and forgot to exhibit a couple of essential documents, such as the certificate from a village panchayat. To make things worse, the lawyer also failed to sufficiently explain the linkages between the documents presented to untangle all the complexities in the case. Momiran Nessa, a middle-aged Bengali-speaking Muslim woman, also from the Barpeta District, had to spend ten years in the Kokrajhar jail before she was granted bail in 2019 after the Supreme Court issued an order to release all alleged migrants who had spent at least three years in a detention centre. As Momiran Nessa's husband died while she was in the detention centre and her children were too young to help, the whole burden of fighting the case fell on her brother, a daily wage labourer who sold the little property he had in order to do so. The first lawyer they hired took a lot of money but turned out to be incompetent, so they had to find a new one—and that too in the later phase of the legal fight. Shahera Khatun, who was false-

ly declared a foreigner and stayed for years in detention and then released after proving her citizenship, spent almost 400,000 rupees on her own case, including the repayment of a high-interest loan. Moslem Ali, whose wife Mozira Khatun is currently in a detention centre, shared his resentment that her lawyer keeps asking for more and more money despite having received 50,000 rupees already and showing no signs of being able to bring her back. Instead, the lawyer often talks about the difficulties of fighting the case, and he has missed all the deadlines he has set for himself to get bail for Moslem Ali's wife. Abdul Salam, a D voter fighting to prove his citizenship, was asked for 50,000 rupees from a lawyer to take his case. He managed to pay 20,000 rupees two years ago, 25, but the case has not yet been resolved. He believes that the lawyer is not helping him because he did not pay the full amount that was demanded (Abdul Salam, excerpt from interview, 2021). While the poverty and general precariousness of suspected migrants make it difficult for them to challenge this status in court, judges rely heavily on the conclusions of the Foreigners Tribunals in deciding whether to detain and 'deport' people. But the tribunals' conclusions are at times very arbitrary, religiously prejudiced and rarely sensitive to the precarious lives of the petitioners.

9. **Illegalisation and Suicide Cases**

The most devastating consequence of the migrant 'illegality' in Assam is the large number of suicide cases reported over the past two years. This includes suicide of the detained inside the detention centre, family members of detainees committing suicide and those who are left out of NRC ending their lives. Ajbahar Ali, an elderly man who was detained and released in 2019, talked about how he lost his wife.

After I lost my case at the high court, I was supposed to file a case in the Supreme Court. For that purpose, I needed about one lakh rupees. We didn't have much savings to meet this. She went into depression and decided to kill herself. After her attempt to end her life, she was taken to the Barpeta Hospital, where she took her last breath. I was not informed about her death for so many days. My family members thought that I would lose my mind after hearing the news (Ajbahar Ali, excerpt from interview, 2021).

A large number of suicide cases have been reported after the publication of the NRC list, which includes a fourteen-year-old girl and an eighty-four-year-old man. Exclusion from the NRC list is not merely about losing their rights as citizens. It profoundly shakes their dignity as human beings. The undoing/unmaking of citizenship leads to the withdrawal from the community and social world to 26, which they have been deeply associated. Many fall into depression and choose to end their life. The fear of detention and deportation also forced many to take the extreme step of suicide. In some instances, suicide comes as a last resort after exhausting all legal battles and struggling to procure proper documents. It is not always the case that the exclusion of one person from NRC forces the individual to commit suicide; there are instances of breadwinners losing the confidence to live when more members or their entire family are excluded from the list

## 10. Conclusion

This paper has investigated what constitutes the South Asian reality of 'bare life' under an increasingly anti-migrant regime in India. Building upon insights from recent ethnographic works whose explorations of undocumented and stateless populations reveal 'bare life' to be a contextual, complex and dynamic experience, it has offered a historically anchored ethnography

of Bengali-speaking population in Assam who have encountered state-driven detection, detention and threats of deportability over the past few decades. These are actual Indian citizens with political rights and a stable social world who have been forced to experience 'bare life', unlike the usual case, in which it is the migrants who are pushed into a dark zone by their illicit border crossing and invalid presence, and contra the predominant 'indistinguishability' argument, which assumes that the Bengali-speaking residents in Assam has benefited from 'documentary citizenship' attained through fraudulent means after going to India from Bangladesh. 'Bare life' in Assam has to be seen as a consequence of various ambivalent policies and arbitrary bureaucratic interventions in the past three decades. The experience of 'bare life' is far more complex in this context as there is a gradual process by which individuals are removed from their social works and rendered right-less. This calls attention to the nature of irregular citizenship in India. The study found how the status of individuals can move to multiple levels from citizenship to partial rights like D voters and to the most vulnerable position of 'illegal migrant', contingent on the bureaucratic interventions and specific challenges of individuals. Further, India's citizenship policies have always been entangled with the complex history of family and kinship relations across borders. The essay specifically addressed how migrant 'illegality' is closely interlinked with the unit of family, both as a fundamental unit for the state in defining citizenship and as an institution that undergoes tremendous crisis under migrant 'illegality'. The study found that family units and their history become the ultimate criteria for defining citizenship and belongingness in the policies adopted by the Indian state to define who is an illegal migrant and who is a citizen. The cases presented in this essay reveal how the incongruencies and errors in the paper records of individuals lead to their failure to prove the Legacy Data, a crucial criterion

that shows the relationship with their ancestors who lived in India. As a consequence, many lost their status of citizenship and were exposed to vulnerable conditions, even leading to the extreme scenario of ending up at the detention centre. The narratives of detained people reveal how "illegalised" humans are coercively kept in a world of extreme suffering and violence. While the state's intention is to categorise people as 'illegal migrants' and 'citizens', there can be no such neat separation between the two as far as the social and familial lives of the people are concerned. In the Western context, scholars have shown how inside/outside question becomes much blurred in mixed-status families as the 'outside' looms over the essentially personal aspects of familial life. In the Indian context, mixed families are largely produced due to certain discursive and governmental practices of the state as the process of detection and identification of 'illegal migrants' leads a normal family into a mixed one. Mixed-status families then live in continuous vulnerability and anxiety with the constant fear of being separated, detained and deported. The illegalisation of a population has a profound impact on their familial and intimate life. It irreparably damages family life as national borders are inscribed in the intimate and social realm of their life. Specifically, these vulnerabilities are tougher for women as they have little or no agency, which also shows the gendered impact of migration policies and citizenship laws in India. The release of several detainees in the recent past also illustrates the fluid and increasingly complex situation of family life among the Bengali-speaking population in India, as deportability hinders their struggle to rebuild their family life. This reading of India scenario perceive 'bare life' as multivalent, complex, gradually transiting and entangled with intimate realm of life. The illegalisation of a large number of people by way of excluding them from the recent NRC list along with CAA, which overtly discriminates against Muslims, renders the crisis more massivel

than ever. This also call for our attention to the 'bare life' rooted in a more heightened politics of religion under current political circumstances as the experience of exclusion and violence is more layered among Bengali speaking residents in Assam.

## Acknowledgements

This essay is an outcome of  the Migration Politics residency programme, I would like to thank Evelyn Ersanilli, Darshan Vigneswaran and my co- fellows, Ulrike Bialas and Jagat Sohail for their valuable feedbacks in shaping this paper. I particularly thank Saskia Bonjour, whose critical comments in various phases of writing and persistent support have been very crucial in sharpening this paper. The article has also benefitted immensely from the comments of three external reviewers– Barak Kalir, Radhika Mongia, Meghna Kajla and one reviewer who selected to remain anonymous. I thank Ritapriya Nandy for her help while developing this paper. I also thank the generous fund I received for the project 'Migration and Marginalities: Exploring North East India and beyond' funded by the Institute of Eminence, University of Hyderabad.

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
