# Peer review of "National Borders among Families: Removal and ‘Bare Life’ in India"

_Migration Politics_

## Round 1 · Referee Report · Meghna Kajla · 2024-2-4

Strengths
The article makes a new pathway while explaining the familial and kinship bonds that are impacted due to changing Citizenship laws, the bureaucratic exercise of NRC, and migration in the state of Assam. The ethnographic details and interview excerpts very well complement the concepts used to demonstrate how intimate issues of marriage alliances, education for children of detainees, and their everyday healthcare remain invisible to the police authorities. The article is interesting in conducting ethnography to explore the anthropology of removal and bare life in the South Asian context. It finds that the South Asian state, specifically India endorsed confusing state policies to manage migration and citizenship in Assam and its long bureaucratic process of creating irregular citizenship.
Weaknesses
The author needs to distinguish between types of migrants and citizens in the context of Assam. Why has the author picked up this ethnicity needs to be explained. The Supreme Court orders require citations. While the focus of the article is on the NRC, it outlines that irregular citizenship is due to the internal policies of the state and turbulence in Assam for five decades. Thus, confusing whether bare lives come to appear due to one policy or is a regularity of the modern state.
Report
Yes, I would recommend the article to be published in this journal.
Requested changes
The article needs to situate the Bengali speaking Muslim community in the context of illegal and legal migrants because the concept of 'migrants' in Assam has a layered history. In doing this, it can clarify the suspicion behind this community of the state and distinguish between illegal, legal migrants, and citizens. As of now, they are intermingled. Last, the changing landscape of Indian politics, i.e., the rise in Hindutva nationalism further disturbs the social fabric of Assam and creates a Hindu-Muslim rift that requires attention.

---

## Round 2 · Author Response

Dear Editor,

I have made minor revisions suggested by the reviewers and summarised by the editor.
Thanks and Regards,

Salah Punathil

---

## Round 2 · List of Changes

1. I have added references to all Supreme Court orders
  2. I have given a clarification in footnote 2 and 3 on the confusing use of migrants and citizens and the differentiation- I have used 'Bengali speaking residents in Assam' in most of the places to avoid the binary and neat distinction. There are more footnotes in the paper now.
  3. Regarding the clarification on 'bare life' in the context of India and moving beyond Agamben, I have provided elaboration in page 9, 11 and in the conclusion apart from the existing clarifications in the paper.
  4. In conclusion, I have added a note on Hindu-Muslim differences in the newer context as this theme has to be addressed separately
  5. I have added acknowledgments

---

## Editorial Decision

unknown